# LEARNING CONCEPT-BASED VISUAL CAUSAL TRANSITION AND SYMBOLIC REASONING FOR VISUAL PLANNING

## ABSTRACT

Visual planning simulates how humans make decisions to achieve desired goals in the form of searching for visual causal transitions between an initial visual state and a final visual goal state. It has become increasingly important in egocentric vision with its advantages in guiding agents to perform daily tasks in complex environments. In this paper, we propose an interpretable and generalizable visual planning framework consisting of i) a novel Substitution-based Concept Learner (SCL) that abstracts visual inputs into disentangled concept representations, ii) symbol abstraction and reasoning that performs task planning via the self-learned symbols, and iii) a Visual Causal Transition model (ViCT) that grounds visual causal transitions to semantically similar real-world actions. Given an initial state, we perform goal-conditioned visual planning with a symbolic reasoning method fueled by the learned representations and causal transitions to reach the goal state. To verify the effectiveness of the proposed model, we collect a large-scale visual planning dataset based on AI2-THOR, dubbed as *CCTP*. Extensive experiments on this challenging dataset demonstrate the superior performance of our method in visual task planning. Empirically, we show that our framework can generalize to unseen task trajectories and unseen object categories. We will release our dataset and codes upon acceptance.

## 1 INTRODUCTION

As one of the fundamental abilities of human intelligence, planning is the process of insightfully proposing a sequence of actions to achieve desired goals, which requires the capacity to think ahead, to employ knowledge of causality and the capacity of imagination (Walker & Gopnik, 2013), so as to reason and foresee the proper actions and their consequences on the states for all the intermediate transition steps before finally reaching the goal state. Visual planning simulates this thinking process of sequential causal imagination in the form of searching for visual transitions between an initial visual state and a final visual goal state. With its advantages in guiding agents to perform daily tasks in the first-person view, visual planning has become more and more important in egocentric vision (Gupta et al., 2017). In robotics, visual planning could also save large amounts of workforce from manually designing the required specific goal conditions, action preconditions and effects for robots.

Previous works for visual planning can be roughly categorized into three tracks, *i.e.*, neural-network-based models (Sun et al., 2022; Oh et al., 2015), reinforcement-learning-based models (Rybkin et al., 2021; Ebert et al., 2018) and classic search-based models (Paxton et al., 2019; Liu et al., 2020). Neural-network-based models can be trained in an end-to-end manner, easily adapting to different tasks and domains. This line of works, however, tends to fall short in terms of its interpretability (Gao & Guan, 2023). Reinforcement-learning-based models can perform goal-conditioned decisions, but could suffer from sparse reward, low data efficiency (Ladosz et al., 2022), and low environment and task generalization ability (Packer et al., 2018). Considering these limitations and inspired by human cognition, we conjecture that there exist three key components for visual planning, namely **representation learning, symbolic reasoning, and causal transition modeling**. Representation learning focuses on extracting objects' relevant, dynamic, and goal-oriented attributes. Symbolic reasoning performs action planning at the abstract higher level via self-learned symbols. Causal transition models the visual preconditions and action effects on attribute changes.

At the **perception** level, we propose to learn concept-based disentangled representation, and believe such human-like perception ability to abstract visual concepts from observations is vital for visual causal transition modeling (Zhu et al., 2020a). The reason is that such representation learning could encode images at a higher semantic level than pixels, distinguish different attribute concepts, extract the "essential" factors of variation, increase robustness and interpretability (Suter et al., 2018; Träuble et al., 2021; Adel et al., 2018), and promise compositional generalization to unseen scenarios with fewer examples in zero-shot inference (Atzmon et al., 2020; Träuble et al., 2021; Higgins et al., 2017; Locatello et al., 2020) as well as serve many real-world down-stream tasks such as causal learning (Träuble et al., 2021). At the **reasoning and planning** level, we argue that understanding the atomic causal mechanisms is crucial and inevitable for task planning. Human infants begin to make causal predictions and provide causal explanations for physical phenomena in the world by 2 years of age (Legare et al., 2010; Hickling & Wellman, 2001). Just as human causal cognition understands causality as events based on the forces of actions and their results (Gärdenfors, 2020), the visual causal transition needs to capture the factors of variation in visual observation and anticipate the effects of actions applied to these factors. The understanding and reasoning of the abstract higher-level task planning composed of the lower-level atomic causal transition also have the potential to be more generalizable and interpretable (Edmonds, 2021; Schölkopf, 2022). Thus, we propose a visual causal transition model as well as its abstracted symbolic transition model. The abstracted symbolic transition corresponds to the discrete higher-level task planning, which is more interpretable, more data-efficient, more reliable and robust, easier to generalize, and better for avoiding the problem of "error accumulation" (Garcez et al., 2022). Guided by symbolic transition, the visual transition reconstructs intermediate and final goal images.

Technically, there are **three critical modules** in our visual planning framework. First, a novel concept learner (Sec. 4.1) is learned by switching the latent concept representations of a pair of images with different attribute concepts. Second, a set of state symbols are abstracted from clustering low-level concept token representations (Sec. 4.2). The most efficient symbolic transition path can be found via a Markov Decision Process (MDP). Third, a visual transition model (Sec. 4.3) is proposed to learn the action-induced transition of the changeable attributes given the concept representations of the precondition image; it serves to generate the resulting effect image. To verify the effectiveness of the proposed framework, we collect a large-scale visual planning dataset, which contains a concept learning dataset and a causal planning dataset. Extensive comparison experiments and ablation studies on this dataset demonstrate that our model achieves superior performance in the visual planning task and various forms of generalization tests.

To summarize our **main contributions**: (i) We propose a novel concept-based visual planning framework, which models both discrete symbolic transition and continuous visual transition for efficient path search and intermediate image generations. Comprehensive experiments show that our method achieves superior performances in visual task planning and generalization tests. (ii) In addition to generalizability, our method has better interpretability by generating a causal chain (the action sequences and the intermediate state images) to explicitly demonstrate the transition process to the goal. (iii) We collect a new large-scale visual planning dataset, which can foster concept and task planning in the community.

## 2 RELATED WORK

### 2.1 VISUAL PLANNING

Visual planning is feasible with the learned representation and atomic causal effects. Lin et al. (2022) proposed a method for long-horizon deformable object manipulation tasks from sensory observations, which relies heavily on differentiable physics simulators. Paxton et al. (2019) performed a tree-search-based planning algorithm on the learned world representation after applying high-level actions for visual robot task planning, but they ignored learning disentangled representations. Sun et al. (2022) learned how to plan a goal-directed decision-making procedure from real-world videos, leveraging the structured and plannable latent state and action spaces learned from human instructional videos, but their transformer-based end-to-end model is hard to generalize to unseen planning tasks. Oh et al. (2015) proposed a model based on deep neural networks consisting of encoding, action-conditional transformation, and decoding for video prediction in Atari Games, but they do not abstract symbols for efficient reasoning. Silver et al. (2021) is the most similar to ours, which learned symbolic operators for task and motion planning, but cannot generate intermediate images.

## 2.2 CONCEPT DISENTANGLEMENT

Concept-based disentangled representation learning has emerged as a popular way of extracting human-interpretable representations (Kazhdan et al., 2021). Discrete and semantically-grounded representation is argued to be helpful for human understanding and abstract visual reasoning, enables few-shot or zero-shot learning, facilitates human-machine interaction, and leads to better downstream task performance (Van Steenkiste et al., 2019; Yu et al., 2022). Automatically learning visual concepts from raw images without strong supervision is challenging in AI research. Previous studies tried to learn disentangled concept representation either in a completely unsupervised manner (Chen et al., 2016; Zhu et al., 2020b; Higgins et al., 2016a; Yang et al., 2022; Yu et al., 2021), or via weak supervision and implicit prototype representations (Stammer et al., 2022), or by employing supervision from the linguistic space (Saini et al., 2022; Mao et al., 2019). There have been diverse learning techniques, such as Transformer (Yang et al., 2022), (sequential) variational autoencoder (Zhu et al., 2020b; Higgins et al., 2016a), and information maximizing generative adversarial nets (Chen et al., 2016), etc. Existing techniques have proved successful on objects mostly with limited variation, such as digits, simple geometric objects (Stammer et al., 2022), and faces (Chen et al., 2016). In this work, we propose a variant of (Yang et al., 2022) by imposing more reconstruction constraints, which works very well on more complex household objects (Sec. 3) and benefits for the downstream planning task compared to prior works.

## 2.3 CAUSAL LEARNING AND REASONING

Visual reasoning for human task understanding is one of the essential capabilities of human intelligence, and a big challenge for AI with the difficulty of generating a detailed understanding of situated actions, their dependencies, and causal effects on object states (Jia et al., 2022). Various evaluated state-of-the-art models only thrive on the perception-based descriptive task, but perform poorly on the causal tasks (*i.e.*, explanatory, predictive, and counterfactual tasks), suggesting that a principled approach for causal reasoning should incorporate not only disentangled and semantically grounded visual perception, but also the underlying hierarchical causal relations and dynamics (Yi et al., 2019). Concept-based disentangled representation learning could benefit causal learning by finding a latent space where important factors could be extracted from other confounding factors, thus facilitating the learning of causal effects (Atzmon et al., 2020). Fire & Zhu (2017) built a sequential Causal And-Or Graph (C-AOG) to represent actions and their effects on objects over time. Our work exploits the disentangled concept representation to ground action to their causal effects on object attributes.

## 3 ENVIRONMENT AND DATASET

To facilitate the learning and evaluation of the concept-based visual planning task, we collect a large-scale RGB-D image sequence dataset named *CCTP* (Concept-based Causal Transition Planning) based on AI2-THOR simulator (Kolve et al., 2017). We exclude scene transitions in each task by design to focus more on concept and causal transition learning, *i.e.*, each task is performed on a fixed workbench, although the workbenches and scenes vary from task to task. The frame resolution is $384 \times 256$, which is converted into $256 \times 256$ at the very beginning of our method. The whole dataset consists of a concept learning dataset and a visual causal planning dataset, which we will illustrate in detail below.

### 3.1 CONCEPT LEARNING DATASET

We learn six different kinds of concepts: `TYPE`, `POSITION_X`, `POSITION_Y`, `ROTATION`, `COLOR`, and `SIZE`. `TYPE` refers to the object category. The dataset has eight different types of objects in total, including *Bread*, *Cup*, *Egg*, *Lettuce*, *Plate*, *Tomato*, *Pot*, and *Dyer*, all of which can be manipulated on the workbench. We manually add the `COLOR` concept to the target object by editing the color of the object in its HSV space. This leads to 6 different colors in all for each object, and 20 samples are provided for each color to avoid sample bias. For `SIZE` concept, we rescale each target object to 4 different sizes as its concept set. As for the position, we use `POSITION_X` and `POSITION_Y` to refer to the coordinates along the horizontal X-axis and the vertical Y-axis w.r.t. the workbench surface. We discretize `POSITION_X` with 3 values and `POSITION_Y` with 5. Notably, changes in `POSITION_X` and `POSITION_Y` also cause variant perspectives of an object. For `ROTATION`, we set 0, 90, 180 and 270 rotation degrees for all types of objects. We exhaustively generate all possible target objects with different value combinations of the six concepts, resulting in

234,400 images. Leveraging the masks provided by AI2-THOR, we isolate the foreground images, containing only the target object with a black background. We randomly choose $40\%$ of the concept combinations for training. For each image $X_{0,f}$ in the training set and each concept index $i$, we search for image $X_{1,f}$ within the training set such that $X_{0,f}$ and $X_{1,f}$ differ only in the $i$-th concept. We use such paired images and the corresponding label $i$ for concept learning.

## 3.2 CAUSAL PLANNING DATASET

A causal planning task consists of several steps of state transitions, each caused by an atomic action. We define seven different atomic actions in our dataset, including `move_front`, `move_back`, `move_left`, `move_right`, `rotate_left`, `rotate_right`, and `change_color`. The magnitude of each action is fixed. The target object states (*e.g.*, its color) are randomly initialized in each task from our dataset. The task lengths (*i.e.*, the number of steps for each task) are not fixed. We collect four subsets of tasks, each representing a difficulty level. In the first level, the workbench has no obstacles, and the ground truth actions involve only movements. In the second level, several fixed obstacles appear on the workbench. In the third level, a dyer additionally appears on the workbench and the target object must be moved adjacent to the dyer to change its color if necessary before being moved to the target position. In the fourth level, rotation actions are involved additionally. The action sequence in each task is paired with the corresponding visual observations. Each subset contains 10,000 tasks: 8,000 for training, 1,000 for validation, and 1,000 for testing.

We construct additional generalization test benchmarks based on our collection. We provide 4 levels of **Unseen Object** generalization tests for object-level generalization. We generate 1000 tasks for each level in which the target object types are unseen in the training dataset, including object types of *Cellphone*, *Dish Sponge*, *Saltshaker*, and *Potato*. Additionally, we have testbeds for generalization tests for unseen tasks. The training and testing tasks in the **Unseen Task** dataset have different combinations of action types. For example, the training dataset may include tasks that consist of only `move_left` and `move_front` actions, as well as tasks that consist of only `move_right` and `move_back` actions, while the testing dataset contains tasks from the held-out data with different combinations. **Unseen Task** dataset is limited to the first and the second difficulty levels because limited combinations of actions are not sufficient to accomplish harder tasks.

## 4 METHOD

Given an initial RGB-D state image $X_0$ and a final RGB-D state image $X_T$, our task is to find a valid and efficient state transition path with an inferred sequence of actions $\Gamma = \{a_t\}_{t=1,...,T}$, as well as generating intermediate and final state images $\tilde{X} = \{\tilde{X}_t\}_{t=1,...,T}$. To fulfill this task, we use a concept learner to extract disentangled concept representations for state images, abstract concept symbols for reasoning, and train a visual causal transition model to generate intermediate state images.

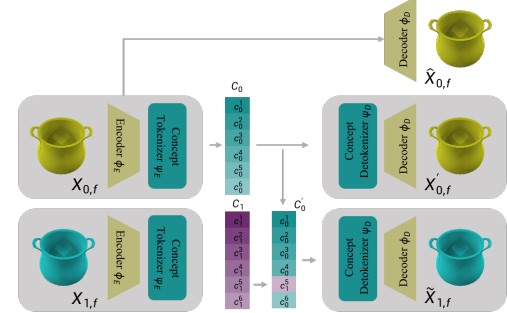

Figure 1: **Architecture of SCL**. Foreground images $X_{0,f}$ and $X_{1,f}$ differ only in the COLOR concept. After extracting their concept tokens, the COLOR concept $c_0^5$ of $X_{0,f}$ is substituted by $c_1^5$ from $X_{1,f}$, which are then fed into the detokenizer and decoder to reconstruct images.

## 4.1 SUBSTITUTION-BASED CONCEPT LEARNER

The architecture of our Substitution-based Concept Learner (SCL) is illustrated in Fig. 1. Given a pair of foreground images $X_{0,f}$ and $X_{1,f}$ as the input, both contain objects that differ only in one concept, *e.g.*, a yellow pot and a green pot. A shared encoder $\phi_E$ is applied to the foreground images to obtain the latent embeddings $Z_{i,f} = \phi_E(X_{i,f})$. The embedding $Z_{i,f}$ is further fed into a concept tokenizer $\psi_T$ to generate the concept tokens $C_i = \{c_i^k\}_{k=1,...,6} = \psi_T(Z_{i,f})$. Here $k$ is the concept index, and we assume there exist six visual concepts, *i.e.*, TYPE, COLOR, SIZE, POSITION_X, POSITION_Y, and ROTATION, representing the visual attributes of the target objects (refer to Sec. 3.1 for details).

The concept token $c_0^i$ is substituted with $c_1^i$ to get a new concept token vector $C_0'$, where $i$ indexes the different concepts between the paired images $X_{0,f}$ and $X_{1,f}$. For example, the token $c_i^5$ represents the color concept in Fig. 1, replacing $c_0^5$ with $c_1^5$ will change the original yellow pot to a green pot. The token vector $C_0'$ is fed into a concept detokenizer $\psi_D$ to reconstruct the latent embedding $Z_{1,f}' = \psi_D(C_0')$, which is further decoded into image $\tilde{X}_{1,f} = \phi_D(Z_{1,f}')$. After the concept detokenizer and decoder, we obtain a combined reconstruction loss as follows:

$$\mathcal{L}_1 = \mathcal{L}_{MSE}(X_{0,f}', X_{0,f}) + \mathcal{L}_{MSE}(\tilde{X}_{1,f}, X_{1,f}), \tag{1}$$

where $\mathcal{L}_{MSE}$ is the mean squared error. In addition, we add another branch directly connecting the encoder to the decoder. This branch aims to distinguish the role of the encoder from that of the concept tokenizer; it enforces the encoder to learn hidden representations by reconstructing $X_{0,f}$. The reconstructed image and reconstruction loss of this branch are $\hat{X}_{0,f}$ and $\mathcal{L}_{MSE}(\hat{X}_{0,f}, X_{0,f})$, respectively. Similar to Yang et al. (2022), a Concept Disentangling Loss (CDL) is employed to reduce interference between the concept tokens. The CDL can be formulated as follows:

$$\mathcal{L}_{CDL} = \mathcal{L}_{CE}(\|C_0 - C_1\|_2, i), \tag{2}$$

where $\mathcal{L}_{CE}$ is the cross-entropy loss. $\|C_0 - C_1\|_2$ calculates the $l_2$ norm of the variation of each concept token. $i$ is the ground-truth token index and indicates that the $i$-th concept token is replaced. The total loss $\mathcal{L}_C$ of concept learner is as follows:

$$\mathcal{L}_C = \mathcal{L}_1 + \mathcal{L}_{MSE}(\hat{X}_{0,f}, X_{0,f}) + \mathcal{L}_{CDL}, \tag{3}$$

where the equal weights for each loss work well in our experimental settings.

## 4.2 SYMBOL ABSTRACTION AND REASONING

Symbol abstraction aims to convert concept tokens into discrete symbols for later symbolic reasoning. Our empirical results in Fig. 6 show that the concept tokens learned in Sec. 4.1 are well-disentangled and can be easily clustered into several categories. Therefore a clustering algorithm could be applied to the concept tokens to generate symbols. Specifically, we collect all the concept tokens extracted from the training data using the substitution-based concept learner and

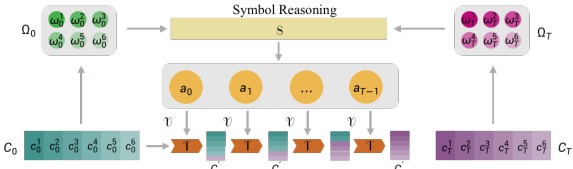

Figure 2: **Symbol abstraction and reasoning**. The symbolic reasoning module generates the most plausible action sequences given the inital and the goal concept symbols. These action sequences are then fed into ViCT to generates effect images.

create the concept token spaces: $\mathbf{C} = \{c_n\}$. Then, we employ the K-means algorithm to cluster data points within the concept spaces, resulting in the concept centers $\{\bar{c}\}$ and a symbol assignment $\omega = \sigma(c, \{\bar{c}\})$ for each concept token $c$. Here $\sigma$ is the nearest neighbor function which assigns the symbol of the nearest concept center to $c$. This process is applied to six defined concepts separately, abstracting a set of concept symbols $\Omega = \{\omega^k\}_{k=1,...,6}$ for each image.

The symbolic reasoning aims to find the most plausible transition path from the initial state to the goal state at the symbol level, which can be formulated as a Markov Decision Process (MDP). Given the initial concept symbols $\Omega_0 = \{\omega_0^k\}_{k=1,...,6}$ and the action $a_0$, the symbol reasoner computes the distribution of concept symbols at the next timestep $\Pr\left[\Omega_1' \mid a_0, \Omega_0\right]$. The concept symbol distribution at the timestep $t$ can be obtained as follows:

$$\Pr\left[\Omega_t' \mid a_{0:t-1}, \Omega_0\right] = \sum_{o \in \boldsymbol{\Omega}} \Pr\left[\Omega_t' \mid a_{t-1}, \Omega_{t-1}' = o\right] \cdot \Pr\left[\Omega_{t-1}' = o \mid a_{0:t-2}, \Omega_0\right], \tag{4}$$

where $\boldsymbol{\Omega}$ denotes the the entire concept symbol space. Additionally, two legality checks are implemented during the reasoning process to ensure the validity of the action sequence, involving action legality and state legality checks. The action legality is defined as $\mathbf{1}_{\Pr[a|\Omega]>\text{thresh}}$. This check aims to prevent the use of noise-inducing transformations caused by the substitution-based concept learner, thereby modifying Equation 4 to:

$$\Pr\left[\Omega_t' \mid a_{0:t-1}, \Omega_0\right] = \sum_{o \in \boldsymbol{\Omega}} \mathbf{1}_{\Pr[a_{t-1}|o]>\text{thresh}} \Pr\left[\Omega_t' \mid a_{t-1}, \Omega_{t-1}' = o\right] \Pr\left[\Omega_{t-1}' = o \mid a_{0:t-2}, \Omega_0\right]. \tag{5}$$

The state legality check is designed to eliminate contributions to the distribution originating from invalid states (e.g., collisions with obstacles on the workbench). It can be written as follows:

$$\Pr\left[\Omega_t^{'} = o_0 \mid a_{0:t-1}, \Omega_0; \{\Omega_{\text{env}}\}\right] = \frac{\mathbf{1}_{o_0 \in \boldsymbol{\Omega}_{\text{valid}}} \cdot \Pr\left[\Omega_t^{'} = o_0 \mid a_{0:t-1}, \Omega_0\right]}{\sum_{o \in \boldsymbol{\Omega}_{\text{valid}}} \Pr\left[\Omega_t^{'} = o \mid a_{0:t-1}, \Omega_0\right]}, \tag{6}$$

where $\boldsymbol{\Omega}_{\text{valid}} \subseteq \boldsymbol{\Omega}$ represents the set of valid concept symbols given the concept symbols of other objects in the environment, and $o_0$ is an arbitrary element within $\boldsymbol{\Omega}$. To reduce computational complexity, the reasoning process is individually applied to each concept. This approach is effective due to the well-designed disentangled concepts, which ensure that the changes in each concept are independent given a particular action. The MDP aims to discover the most possible action sequence $a_{0:T-1}$ for which the corresponding distribution of concept symbols $\Omega_T^{'}$ closely approximates the goal concept symbols $\Omega_T$. This action sequence is then passed into the Visual Causal Transition model (See Sec. 4.3) to generate predicted intermediate images (Fig. 2).

### 4.3 VISUAL CAUSAL TRANSITION LEARNING

The aim of visual causal transition model (ViCT) is to generate visual effect images based on visual precondition images and human actions. For example, Fig. 3 shows an action that moves the pot one step to the right. ViCT predicts the low-position image $X_1$ by transforming the high-position image $X_0$ with a put_down action.

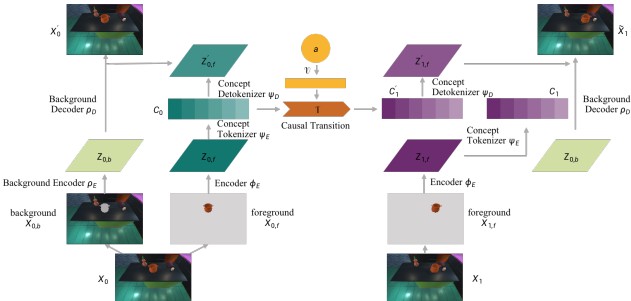

As seen in Fig. 3, three parts exist in the framework of ViCT. Firstly, the causal transition is the key part of ViCT. This process transforms object concept tokens from $C_0$ to $C_1^{'}$ with the help of an action embedding $\mathcal{V}(a)$. The action $a$ is encoded into a one-hot vector and further embedded via an embedding function $\mathcal{V}$ to achieve this. The transition process is as follows:

Figure 3: **Architecture of ViCT**. The concept tokenizer extracts object concept tokens for causal transition. The causal transition model transforms concept tokens from $C_0$ to $C_1^{'}$ with the action embedding $\mathcal{V}(a)$. The background encoder converts the background image into latent vectors, which are then combined with predicted concept tokens $C_1^{'}$ to generate the effect image $\tilde{X}_1$.

$$C_1^{'} = \mathcal{T}(C_0, \mathcal{V}(a)), \tag{7}$$

where $C_1^{'}$ represents the resulting concept tokens. $\mathcal{T}$ denotes the transition function involved in this causal transition process.

In addition to the causal transition component, two other crucial parts in ViCT are dedicated to managing visual extraction and reconstruction. The second part contains a concept tokenizer to extract foreground object concept tokens $C_0$ for later transitions. This concept tokenizer has been trained as described in Sec. 4.1 and fixed here. This part also involves a background encoder $\rho_E$, which processes the background image to produce latent vectors represented as $Z_{0,b}$. The vectors $Z_{0,b}$ store background-related information and will be used to generate the resultant image $\tilde{X}_1$, as illustrated in the rightmost part of Fig. 3. The third part combines foreground object concept tokens and background latent vectors to predict effect image $\tilde{X}_1$ with the background decoder $\rho_D$. Instead of directly using concept tokens, we convert them back to latent embeddings, *i.e.*, from $C_1^{'}$ to $Z_{1,f}^{'}$, and then concatenate $Z_{1,f}^{'}$ with latent vectors $Z_{0,b}$ as the input to the decoder. Similarly, we can also combine $Z_{0,f}^{'}$ and $Z_{0,b}$ to obtain a reconstruction image $X_0^{'}$.

Up to now, two losses can be computed during training: a reconstruction loss $\mathcal{L}_{MSE}(X_0^{'}, X_0)$ and a prediction loss $\mathcal{L}_{MSE}(\tilde{X}_1, X_1)$. In addition to measuring image-level prediction errors, we can also evaluate token-level prediction errors. Given a ground-truth effect image $X_1$, we extract its concept tokens $C_1$, and introduce a token prediction loss $\mathcal{L}_{MSE}(C_1^{'}, C_1)$.

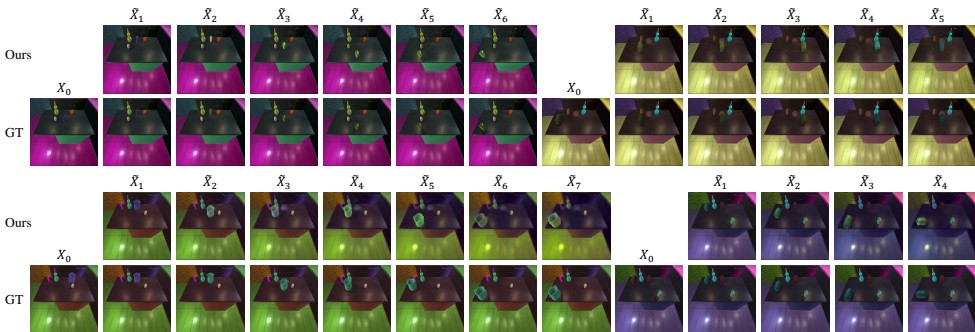

Figure 4: **Qualitative results of our visual planning model**. The top two samples are obtained from the level-3 dataset, and the bottom two are from the level-4 datasets. Our model demonstrates its capability to manage tasks of varying lengths, effectively planning action sequences, and generating intermediate and goal state images. Notably, the first sample from the level-4 dataset generates a different path compared to the ground truth but still achieves success and maintains high efficiency.

The total loss of ViCT is summarized as follows:

$$\mathcal{L}_T = \mathcal{L}_{MSE}(C_1', C_1) + \mathcal{L}_{MSE}(\tilde{X}_1, X_1) + \mathcal{L}_{MSE}(X_0', X_0). \tag{8}$$

The visual causal transition model is trained on our causal planning dataset (see Sec. 3.2).

## 5 EXPERIMENTS

In our experiments, we aim to answer the following questions: (1) Is our model design effective and applicable to visual planning tasks? (2) How do the proposed key components contribute to the model performance? (3) Are the learned concepts and causal transitions interpretable? (4) Does the proposed method exhibit generalization on novel tasks? To answer these questions, we perform extensive experiments on dataset *CCTP*. As shown, the proposed methods are interpretable, generalizable, and capable of producing significantly better results than baseline methods.

### 5.1 EVALUATING VISUAL PLANNING ON DATASET *CCTP*

To validate the effectiveness of our model design, we employ PlaTe (Sun et al., 2022), the state-of-the-art method for visually-grounded planning, as our baseline. To probe the contribution of our proposed components, we replace each component with alternative baselines to compare with. We replace the proposed concept learner with strong baselines such as beta-VAE (Higgins et al., 2016b) and VCT (Yang et al., 2022) model to verify the effectiveness of our concept learning module. Additionally, we compare our model to a reinforcement-learning-based decision process, noted as "w/ RL". Furthermore, to verify the necessity of our symbolization process, we apply the reasoning process directly to the concept tokens, employing our causal transition model to search for states closest to the goal state within the concept token spaces. We also conduct experiments where we further remove the concept learning process. Instead, we use an autoencoder

Table 1: **Quantitative results for visual task planning.** Models corresponding to the model IDs are: 1. Chance, 2. PlaTe (Sun et al., 2022), 3. Ours w/ $\beta$-VAE (Higgins et al., 2016b) , 4. Ours w/ VCT (Yang et al., 2022), 5. Ours w/o symbol, 6. Ours w/o concept, 7. Ours w/o causal, 8. Ours w/ RL, 9. **Ours**. The best scores are marked in **bold**.

| Model ID | ASAcc.(%)(↑) Top-1 | Top-5 | ASE(↑) | FSD(↓) | ASAcc.(%)(↑) Top-1 | Top-5 | ASE(↑) | FSD(↓) |
|---|---|---|---|---|---|---|---|---|
| | Dataset level-1 | | | | Dataset level-2 | | | |
| 1 | 1.3 | 7.3 | - | 3.139 | 0.4 | 2.2 | - | 3.499 |
| 2 | 38.9 | - | - | - | 15.3 | - | - | - |
| 3 | 0.5 | 3.0 | 0.970 | 3.220 | 0.0 | 3.5 | - | 3.670 |
| 4 | 54.1 | 60.6 | 0.972 | 1.483 | 1.6 | 4.9 | 0.988 | 1.294 |
| 5 | 65.8 | 76.9 | 0.983 | 1.197 | 41.0 | 52.6 | 0.962 | 1.627 |
| 6 | 56.9 | 77.6 | 0.986 | 1.644 | - | - | - | - |
| 7 | 1.4 | - | - | 3.326 | 0.3 | - | - | 3.419 |
| 8 | 29.7 | 35.1 | **0.991** | 2.418 | 2.5 | 6.0 | **1.000** | 3.150 |
| 9 | **97.9** | **99.2** | 0.971 | **0.025** | **99.4** | **99.6** | 0.981 | **0.013** |
| | Dataset level-3 | | | | Dataset level-4 | | | |
| 1 | 0.0 | 0.4 | - | 3.513 | 0.1 | 0.4 | - | 3.147 |
| 2 | 0.7 | - | - | - | 0.4 | - | - | - |
| 3 | 0.0 | 0.5 | - | 3.596 | 0.0 | 0.0 | - | 3.107 |
| 4 | 0.7 | 1.2 | 0.968 | 3.442 | 0.2 | 0.3 | 1.000 | 3.193 |
| 5 | 15.4 | 24.1 | 0.970 | 2.278 | 9.8 | 14.0 | 0.981 | 2.149 |
| 7 | 0.0 | - | - | 3.691 | 0.0 | - | - | 3.201 |
| 8 | 3.0 | 3.9 | **1.000** | 3.030 | 2.8 | 3.5 | **1.000** | 2.498 |
| 9 | **86.5** | **87.0** | 0.966 | **0.037** | **55.1** | **76.7** | 0.978 | **0.003** |

to extract latent embedding for causal transition. The corresponding results are denoted as "w/o. symbol" and "w/o. concept", respectively. The "w/o. concept" experiments are limited to the level-1 dataset because the method is unable to handle obstacles. Finally, we replace the explicit planning

module with a transformer architecture. It takes the concept symbols of the initial state and goal state, provided by our concept learner and symbolizer, as inputs to generate the action sequence. We refer to this variant as "w/o causal". We also substitute the planning module with random action predictions for each step as an additional baseline for reference. Detailed implementation for baselines is illustrated in Sec. A.3.

**Evaluation metrics**    To thoroughly inspect the performance of visual planning, we employ metrics including Action Sequence Prediction Accuracy (ASAcc), Action Sequence Efficiency (ASE), and Final State Distance (FSD). ASAcc evaluates the sequence prediction accuracy. In level-1 and 2 tasks, a successful prediction entails moving the target object accurately to the position of goal states without encountering any collisions with obstacles (if present). In level-3 tasks, when the target object's color changes, success requires moving the object adjacent to the dyer, applying the change_color action, and then moving it to the goal position. In level-4 tasks, the target object must also be correctly rotated for success. ASacc is measured as the success rate. During testing, the planning models make 5 attempts for each task. The top-1 accuracy is based on the first attempt, while the top-5 accuracy checks if any of the 5 attempts are successful. ASE measures the efficiency of the planning by comparing the length of the ground truth sequence to that of the predicted sequence. We only take the successfully predicted sequences into consideration. The ASE is defined as follows:

$$ASE = \frac{\sum_{i=1}^{N} \mathbb{I}(\mathbf{\Gamma}_i^{pred})\ell(\mathbf{\Gamma}_i^{gt})/\ell(\mathbf{\Gamma}_i^{pred})}{\sum_{i=1}^{N} \mathbb{I}(\mathbf{\Gamma}_i^{pred})}, \tag{9}$$

where $\mathbb{I}$ is a indicator function for a successful prediction, $\ell$ represents the length of an action sequence. Of note, the ground truth action sequences in *CCTP* are the most efficient, so the efficiency of a predicted sequence will be no more than 1. FSD calculates the distance between the positions of the foreground object in the final predicted state and in the goal state. The distance is defined based on the object's coordinates w.r.t. the workbench.

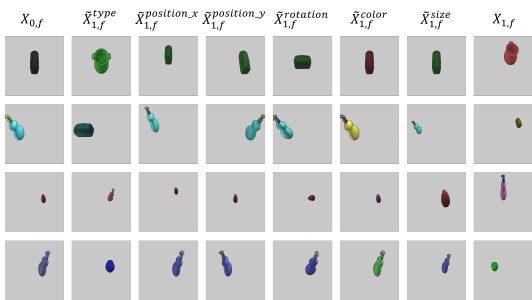

Figure 5: **Fine-grained attribute level concept manipulation.** The concept learner generates new images by substituting each concept token $c_0^i$ from $X_{0,f}$ with $c_1^i$ from $X_{1,f}$.

**Results**    We can see from Tab. 1 that the proposed method achieves significantly higher performance compared with baselines. Specifically, we compare our method with different ablative variants and a strong baseline PlaTe (Sun et al., 2022). Our method outperforms baselines in terms of sequence accuracy (ASAcc) by a large margin and achieves the smallest final state distance (FSD), which demonstrates our method can obtain an accurate planning path to reach the goal state. Our method achieves very competitive ASE if not the best among all the models. Moreover, our model maintains strong performance when encountering hard tasks, while competitive baselines' performances significantly decrease as task difficulty increases. These results demonstrate the effectiveness of our model design. Our full model achieves the best overall performance in all four levels of tests, and each component of our model contributes remarkably to the performance improvements. Of note, our experiments demonstrate a large boosted performance by adding symbolic transition. The qualitative results are shown in Fig. 4.

## 5.2 INTERPRETABLE CONCEPTS AND CAUSAL TRANSITIONS

We qualitatively show the interpretability of the concept learned by our model. We randomly choose 2 images $X_{0,f}$ and $X_{1,f}$, substituting the concept token $c_0^i$ with $c_1^i$ for $i = 1, 2, 3, 4, 5, 6$, which are then fed into the concept detokenizer and the decoder to generate new images. As Fig. 5 shows, with the properly learned concept representations, we could perform fine-grained attribute-level concept manipulation. This indicates that our concept learner is capable of disentangling concept factors and demonstrates the interpretability of our method.

We quantitatively demonstrate the interpretability of our learned causal transitions with statistics of the corresponding causal effects. To be specific, we aim to answer the question: do the learned causal transitions have semantic meaning consistent with the corresponding action? Fig. 6 (a) shows the correlation between concepts and actions, measured with $l_2$ norm between the concept vectors before and after each action. A larger $l_2$ norm means a higher correlation. We can see that the

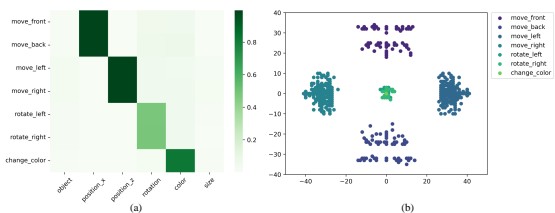

Figure 6: **Action effects on the learned disentangled concept representations.** (a) $l_2$ norm between the concept vectors before and after each action. (b) Distributions of position change induced by each action.

learned rotation actions only affect the rotation status in the concept vector. Similarly, the horizontal and vertical movements only affect the x and y coordinates. Fig. 6 (b) shows the distribution of position change induced by 7 displacement actions. For example, the position changes of `move_front` distribute along the positive y-axis, while those of `move_back` distribute along the negative y-axis. This evidence indicates that 1) our learned concept is successfully disentangled, which makes it possible for our model to learn causal transitions, and 2) the learned causal transition is consistently grounded to real-world actions with similar semantics.

## 5.3 GENERALIZATION ON NOVEL OBJECTS AND TASKS

We design two experiments to test the generalizabiliy of our model.

**Unseen objects** Through this experiment, we aim to investigate if our model can perform visual planning tasks on objects unseen during training. We test our model on the **Unseen Object** testing dataset (see Sec. 3.2 for details) and compare the results with several baselines to demonstrate the generalizability of our concept-based object representation module. We expect our concept learner to recognize the color, position, and size attributes of unseen object types during testing. If this is the case, the transition model could consequently apply transitions on these visual attributes for successful manipulation tasks. As shown in Tab. 2, our model is significantly more robust than PlaTe and RL-based methods against novel objects.

Table 2: **Quantitative results for generalization tests.** Models corresponding to the model IDs are: 1. Chance, 2. PlaTe (Sun et al., 2022), 3. Ours w/o symbol, 4. Ours w/o concept, 5. Ours w/o causal, 6. Ours w/ RL, 7. **Ours**. The best scores are marked in **bold**.

| Model ID | ASAcc.(%)(↑) | | ASE(↑) | FSD(↓) | ASAcc.(%)(↑) | | ASE(↑) | FSD(↓) |
|---|---|---|---|---|---|---|---|---|
| | Top-1 | Top-5 | | | Top-1 | Top-5 | | |
| | **Unseen Object** level-1 | | | | **Unseen Object** level-2 | | | |
| 1 | 0.6 | 4.7 | - | 3.203 | 1.1 | 3.2 | - | 3.591 |
| 2 | 18.5 | - | - | - | 9.7 | - | - | - |
| 3 | 44.0 | 59.9 | 0.968 | 1.507 | 29.0 | 43.8 | 0.986 | 1.880 |
| 4 | 37.1 | 60.5 | 0.950 | 1.319 | - | - | - | - |
| 5 | 1.7 | - | - | 3.233 | 0.2 | - | - | 3.563 |
| 6 | 30.2 | 35.9 | **0.989** | 1.887 | 2.2 | 6.1 | **1.000** | 3.549 |
| 7 | **72.4** | **97.2** | 0.987 | **0.470** | **73.2** | **93.6** | 0.978 | **0.491** |
| | **Unseen Object** level-3 | | | | **Unseen Object** level-4 | | | |
| 1 | 0.0 | 0.0 | - | 3.544 | 0 | 0.1 | - | 3.518 |
| 2 | 0.6 | - | - | - | 0.8 | - | - | - |
| 3 | 12.6 | 22.5 | 0.990 | 2.710 | 6.9 | 11.7 | 0.972 | 2.917 |
| 5 | 0.0 | - | - | 3.467 | 0.0 | - | - | 3.183 |
| 6 | 1.9 | 5.3 | **1.000** | 3.484 | 1.4 | 4.9 | **1.000** | 3.370 |
| 7 | **61.8** | **66.9** | 0.960 | **0.307** | **29.1** | **43.9** | 0.954 | **0.424** |
| | **Unseen Task** level-1 | | | | **Unseen Task** level-2 | | | |
| 1 | 0.4 | 2.1 | - | 3.550 | 0.1 | 0.3 | - | 3.513 |
| 2 | 1.4 | - | - | - | 0.5 | - | - | - |
| 3 | 63.1 | 78.0 | 0.974 | 1.022 | 40.0 | 51.9 | 0.980 | 1.407 |
| 4 | 42.7 | 70.7 | 0.971 | 1.485 | - | - | - | - |
| 5 | 0.0 | - | - | 3.536 | 0.0 | - | - | 3.525 |
| 6 | 26.3 | 30.1 | **0.994** | 2.159 | 2.8 | 7.0 | **1.000** | 3.417 |
| 7 | **98.7** | **99.3** | 0.985 | **0.015** | **98.2** | **99.4** | 0.991 | **0.019** |

**Unseen tasks** Moreover, we aim to verify that our model is flexible in processing atomic actions. We train our model on tasks with only limited types of action combinations, *i.e.*, the **Unseen Task** dataset. In this experiment, PlaTe only performs at the same level as a random guess, while our model performs as well as it does when being trained on the whole dataset, which demonstrates the generalizability of our method on unseen tasks.

## 6 CONCLUSION

In this paper, we propose a novel visual planning model based on concept-based disentangled representation learning, symbolic reasoning, and visual causal transition modeling. In the future, we plan to extend our model to real world task planning, particularly to robotic manipulation.

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

## APPENDIX A    MODEL IMPLEMENTATION DETAILS

### A.1    SUBSTITUTION-BASED CONCEPT LEARNER

For $256 \times 256$ images, the encoder transforms them into $8 \times 8$ with 64 channels by a sequence of networks: 3 convolutional layers and 2 residual blocks, followed by another 4 convolutional layers and 2 residual blocks. The decoder involves 4 transposed convolutional layers and 2 residual blocks, followed by another 3 transposed convolutional layers and 2 residual blocks. We use the architecture in VCT (Yang et al., 2022) as our concept tokenizer and detokenizer. The concept number is set to 6.

### A.2    SYMBOL REASONING AND VISUAL CAUSAL TRANSITION

In the causal transition model, the action is initially embedded into a 64-dimensional vector, which is then concatenated with six separate 64-dimensional concept vectors. Following this, a four-layer MLP is applied for each concatenated vector to predict the six affected concept vectors.

The symbol-level transition model logs all the (input, action, output) triplets in the training data. It forecasts the probability distribution for all six affected concepts when provided with the input concept symbol and the action.

In the visual extraction process, the background encoder involves 3 convolutional layers and 2 residual blocks, transforming background images into $64 \times 64$ latent vectors with 64 channels. The background decoder involves two decoding modules and a transition module. The first decoding module has 4 convolutional layers and 2 residual blocks, decoding the front latent vectors into $64 \times 64$ with 64 channels. Then the transition module is applied, involving 3 convolutional layers and 3 transposed convolutional layers, converting the concatenated front and background vectors into the transitioned background latent vectors, which are $64 \times 64$ with 64 channels. After that, the transitioned background latent vectors are concatenated with the front latent vectors again, being fed into the second decoder module, which involves 3 transposed convolutional layers and 2 residual blocks and decodes the vectors into effect images.

During training, all models are optimized by Adam (Kingma & Ba, 2014), with the start learning rate $10^{-5}$ for concept tokenizer and detokenizer and $3 \times 10^{-4}$ for the rest models. We train our SCL for 180 epochs and our ViCT model for 70 epochs on a single NVIDIA RTX 3090 GPU.

### A.3    BASELINES

**Reasoning applied on the continuous domains**    In Sec. 5.3, we introduced two baselines that involve a reasoning process within continuous vectors. The "w/o. symbol" method employs our trained causal transition model to explore the action space, aiming to discover the action sequence that transforms the concept tokens closest to the goal state. The distance between concept tokens is measured using the l2-norm. Similarly, a causal transition model based on image vectors is trained for the "w/o. concept" method in later experiments. One significant drawback of these methods is that defining action validity becomes challenging. This implies that these methods might predict action sequences that move the target object outside the workbench, resulting in vectors that the transition model cannot comprehend.

**Reinforcement Learning**    A goal-conditioned Double DQN agent (Van Hasselt et al., 2016) is trained with prioritized experience replay (Schaul et al., 2015) to choose actions, taking the concept symbols of the current state and goal state given by our concept learner as inputs. We use our symbolic reasoning model to mimic the learning environment for the agent. The symbolic reasoning model applies the chosen action to the concept symbols and returns the concept symbols of the next state for the agent. The agent gets a reward of 1 only if the current concept symbols are equal to the goal state's symbols.

**PlaTe**    PlaTe (Sun et al., 2022) is a Transformer-based planning method that simultaneously learns an action predictor based on current and goal state features and a state predictor based on the predicted action and state features. We follow the official implementation of PlaTe and use a pre-trained

ResNet-50 (He et al., 2016) to extract 1024-d features of images of the target object, dyer, and obstacles separately as the input state features. We use the features of one image frame as one state. The model is trained without parameter tuning for 500 epochs on dataset *CCTP*.

**VCT**   VCT (Yang et al., 2022) is an unsupervised method to extract disentangled concepts from simple images. We trained a VCT model with our image encoder and decoder architectures on our concept dataset. It is capable of reconstructing the images well but fails to achieve disentanglement.

