# OpenReview forum: "Learning Concept-Based Visual Causal Transition and Symbolic Reasoning for Visual Planning"
_ICLR.cc/2024/Conference — ICLR 2024 Conference Withdrawn Submission_

### Official Review · Reviewer_h59h · 2023-10-13

**Soundness:** 2 fair
**Presentation:** 3 good
**Contribution:** 1 poor
**Rating:** 3
**Confidence:** 4

**Summary:**

This paper proposes a visual planning method, which proposes a concept learning module, a symbolic abstract module, and a visual causal transition model. The resulting model demonstrated superior performance in a visual planning dataset called CCTP. Besides, the proposed framework can generalize to unseen task trajectories and unseen object categories.
---
Post rebuttal comments: it seems that this paper did not submit a rebuttal, so the reviewer holds his score unchanged.

**Strengths:**

1. The idea is interesting and sound. It mainly shows that a neural-symbolic method can have better interpretability and generalization performance.
2. This paper is well-written. Although the model is complex, this paper flows well. It means that the authors may spend a lot of time polishing this paper.
3. The ablation studies are extensive.
4. The related papers are adequately referenced.

**Weaknesses:**

# 1. Motivations of the dataset are not enough:

(1) The dataset mainly contains objects of low-level attributes TYPE, COLOR, SIZE, POSITION X, POSITION Y, and ROTATION. This is not interesting enough and I believe in such a simple dataset, end-to-end RL approaches will be better.

(2) The baseline PlaTe dataset looks even more interesting as it involves cutting, mixing, or other actions. Why don't the authors use existing datasets?

(3) The dataset is toy and synthetic, however, nowadays there are many large-scale video reasoning datasets, such as [1][2][3]. I wish to warn the authors that in 2023 and future years, synthetic datasets will not be encouraged especially they are designed for some special purposes. In previous years we saw many papers in this domain using synthetic datasets, this was because at that time we could not control realitic datasets easily and these papers did include a lot of novelty in concepts or insights.

(4) The method seems to be too complex to overcome such a simple dataset.

(5) The proposed dataset looks similar to an old paper [4], since [4] is also a neural-symbolic approach, it's better to connect to their papers. Also, I don't think the proposed dataset is superior to [4].

# 2. Comparisons are far from enough:

(1) The authors claim the previous visual planning work "fall short in terms of its interpretability". However, I do not see any convincing comparisons in interpretability in this paper. This claim is not supported by experiments. Fig 5 shows some visualizations, but I'm confused by the figure, what does each row mean and what does each column mean? What do the symbols refer to in this diagram?

(2) Moreover, I don't think a very complex method will lead to stronger interpretability, which is counterintuitive. If a method involves so many modules, the interpretability will decrease naturally.

(3) The visualization in Fig 5 cannot convince me because the authors may cherry-pick their results to show good "interpretability". Normally from my experiments in neural-symbolic learning, those concept modules can also produce wrong results and it is hard to say which part is wrong. More theoretical results or more statistical results are required to demonstrate the interpretability or safety.

(4) The authors should compare to previous visual planning work such as Rybkin et al., 2021; Ebert et al., 2018 in the paper. Trust me, I believe in such a simple environment, end-to-end approaches work the best.

# 3. About the novelty:

I don't think the proposed model is particularly novel in the context of neural-symbolic learning. However, I wish the authors not to be discouraged by this fact and the authors are suggested to pick a good challenging environment first. The first limitation I mentioned is the most important.

(1) There are too many papers using this framework like representation learning, symbolic reasoning, and causal transition modeling. For example, NSCL, [4], and [5] are all similar to the proposed framework. I don't think there is much novelty in the framework.

(2) No particularly novel modules are produced in this work and there is not much conceptual novelty. This paper also does not show very amazing empirical results.

[1] STAR: A Benchmark for Situated Reasoning in Real-World Videos
[2] CLEVRER: CoLlision Events for Video REpresentation and Reasoning
[3] https://paperswithcode.com/dataset/something-something-v2
[4] NEURAL PROGRAMMER-INTERPRETERS
[5] Visual Programming: Compositional visual reasoning without training

**Questions:**

I think the authors can refer to the previous section because I talked a lot about the suggestions, settings, methods, and novelty. I warmly suggest the authors to refer to rejected papers in previous ICLR conferences in neural-symbolic AI. This can help the authors reduce cycles needed to publish their paper.

---

### Official Review · Reviewer_mQyd · 2023-10-15

**Soundness:** 2 fair
**Presentation:** 3 good
**Contribution:** 2 fair
**Rating:** 5
**Confidence:** 2

**Summary:**

The paper solves the visual planning task, which aims to search for visual causal transitions between an initial visual state
and a final visual goal state.
It proposes an interpretable and generalizable visual planning framework, equipped with a concept learner, symbol abstraction reasoning, and a visual causal transition model.
The effectiveness of the proposed method is verified by a new large-scale visual planning dataset CCTP, which is collected based on AI2-THOR.

**Strengths:**

1. The paper has an interesting motivation to integrate representation learning, symbolic reasoning, and causal transition modeling into the visual planning task.
2. The paper makes some technical contributions, including a novel Substitution-based Concept Learner and a Visual Causal Transition model.
3. The paper collects a new large-scale visual planning dataset CCTP, which can foster concept and task planning in the community.

**Weaknesses:**

1. Evaluation is limited.
- The experiment is conducted only on the newly collected CCTP dataset. However, there are some other benchmarks, e.g., CrossTask used in PlaTe.
- There is only one sota baseline PlaTe. Setting 3-8 in Tab.1&2 are actually ablation studies. This makes it difficult to judge the effectiveness of the proposed method.
- The CCTP benchmark is a synthetic benchmark with simplified tasks, e.g., move_left. I am concerned about the real-world application of the proposed method. For example, in CrossTask benchmarks used in PlaTe, the procedure planning for "Make Pancakes" cannot expressed by the simple concepts (position, size, .etc) defined in the paper.

2. The definition of the concepts and planning tasks needs consideration.
- As mentioned above, these simple concepts cannot solve real-world complex tasks.
- Also, it is not clear how some of the tasks is related to practical application. For example, the action "change color" should be connected real-world operation such as painting, making it not as a easy task as "move right".
- It is not clear how some of the defined concepts (e.g., "size") related to the defined planning task.

**Questions:**

See in "Weaknesses"

---

### Official Review · Reviewer_YcnS · 2023-11-01

**Soundness:** 3 good
**Presentation:** 3 good
**Contribution:** 2 fair
**Rating:** 3
**Confidence:** 4

**Summary:**

This paper presents a multi-step solution to visual planning various low-level actions in a simulation environment. The authors emphasize decoupling various concepts such as color, type, and position, and then planning in a simpler space than a visual one. The authors also propose a new dataset in order to test visual planning systems which includes substitutions of those concepts in order to directly encourage covariance with the features.

**Strengths:**

+ The paper proposes a nice breakdown of individual problems in order to solve the larger task in stages.
+ The authors create a novel dataset that could be useful to tease out methods, especially those that claim decoupling of various properties
+ The approach was mostly well explained, and diagrams improved my understanding of the architectures

**Weaknesses:**

- I thought the problem setup was a little simplistic. While I don't mind the actions being somewhat discrete, having the states be so discrete makes me think you could essentially memorize or template match the various concepts and skip a whole lot of the learned approach here. This was my biggest concern with the paper.
- I am unclear on the loss used to learn the planning system, and as such it feels like actions are invented from thin air. It is unclear if the algorithm is supposed to plan multiple steps, and if so how it has a loss signal along the whole plan. The Symbol Reasoning architecture and loss setup could have significantly more detail. Given that this is a key component to the system, it should be explained more directly.
- I found many of the diagrams a little small, but especially the reconstruction results. With results so small, it's hard to tell what are compression artifacts vs what are inaccuracies of the network.
- Some statements were so vague as to become completely useless. "we compare our model to a reinforcement-learning-based decision process, noted as “w/ RL”" RL is an entire field of machine learning, how is a reader to understand how RL is applied here, what learning procedures are used etc.
- The tables are very hard to parse with the method names above the table rather than inside it. If these are the most important results of the paper, they should warrant the space needed to list the methods
- I am somewhat suspicious of the poor performance of the only baseline model used (Sun et. al.). The original model was presented on a much more open-ended dataset with real-world images. On this simpler problem, it should do reasonably well. The authors offer no explanation on the failings of the PlaTe model, nor do they even offer comparable Top 5, ASE, or FSD numbers. Rather than compare PlaTe on the new dataset (or in addition), I would have liked to see the authors apply their model to the dataset used in Sun et. al. If the authors can show improvement over PlaTe on that data, I think it would be a much stronger paper.

**Questions:**

Minor comment, not a pro or a con, just a nit: Top 1 and Top 5 are not really the right names. They should be 1 try vs 5 tries because they are not ranked in any way.

I would like the authors to provide more detail on the RL model used in their benchmark.
I would like more explanation to the Symbol Reasoning architecture and how it handles time.
If possible I would like more of an exploration of why PlaTe did so poorly on the dataset, and a comparison of the proposed model on the Sun et al. data.

---

### Official Review · Reviewer_g4Qf · 2023-11-05

**Soundness:** 3 good
**Presentation:** 3 good
**Contribution:** 3 good
**Rating:** 6
**Confidence:** 3

**Summary:**

This paper presents a framework for visual planning, which simulates the decision-making process humans use to achieve desired goals using visual and symbolic reasoning. This paper introduces a novel Substitution-based Concept Learner (SCL) that extracts visual concepts of an image and a generative Visual Causal Transition model (VCT) that guides visual symbol abstraction and reasoning for task planning through learned visual symbols. The model is designed to handle both goal-oriented visual planning and symbolic reasoning in a structured, generalizable fashion. The experiments demonstrate that their approach can generalize well to unseen tasks and object categories.

**Strengths:**

1. The paper proposes a concept-based visual planning work, which demonstrate its efficacy through through experiments, with a clear ablation for each module.
2. The proposed model show consistent robustness to recognize the color, size and position attributes of unseen objects, with a great performance gain over the baseline PlaTe and RL-based methods. The generalizability is also observed in unseen tasks setting.\
3. A large-scale causal planning dataset is also collected which may benefit future research work in the visual planning domain.

**Weaknesses:**

1. The clarity of the paper is generally good, especially the approach details are explained in details, thought he presentation of the qualitative results may need a further improvement. E.g., Figure 4 is a little confusing to parse with a first look, besides Fig 5 is not even explained in the main paper, which also make it hard to comprehend.
2. The details analysis of how the proposed model can generalize to unseen objects with various learned concept attributions, and their combinations are underexplored. E.g., the only presented unseen object level-1 level-2 only shows the subjective understanding of how difficult are the new object categories to recognize, but should have been more clear if the numbers can be shown under different concepts attributes/combinations.

**Questions:**

Illustrated in the weaknesses section.